# Promoter nucleosome dynamics regulated by signalling through the CTD code

**Philippe Materne[1†], Jayamani Anandhakumar[1†‡], Valerie Migeot[1], Ignacio Soriano[2], Carlo Yague-Sanz[1], Elena Hidalgo[3], Carole Mignion[1], Luis Quintales[2], Francisco Antequera[2], Damien Hermand[1*]**

[1]URPHYM-GEMO, Namur Research College, University of Namur, Namur, Belgium; [2]Instituto de Biología Funcional y Genómica, Consejo Superior de Investigaciones Científicas, Universidad de Salamanca, Salamanca, Spain; [3]Departament de Ciencies Experimentals i de la Salut, Universitat Pompeu Fabra, Barcelona, Spain

**Abstract** The phosphorylation of the RNA polymerase II C-terminal domain (CTD) plays a key role in delineating transcribed regions within chromatin by recruiting histone methylases and deacetylases. Using genome-wide nucleosome mapping, we show that CTD S2 phosphorylation controls nucleosome dynamics in the promoter of a subset of 324 genes, including the regulators of cell differentiation *ste11* and metabolic adaptation *inv1*. Mechanistic studies on these genes indicate that during gene activation a local increase of phospho-S2 CTD nearby the promoter impairs the phospho-S5 CTD-dependent recruitment of Set1 and the subsequent recruitment of specific HDACs, which leads to nucleosome depletion and efficient transcription. The early increase of phospho-S2 results from the phosphorylation of the CTD S2 kinase Lsk1 by MAP kinase in response to cellular signalling. The artificial tethering of the Lsk1 kinase at the *ste11* promoter is sufficient to activate transcription. Therefore, signalling through the CTD code regulates promoter nucleosomes dynamics.

**\*For correspondence:** Damien. Hermand@unamur.be

†These authors contributed equally to this work

**Present address:** ‡Department of Biochemistry and Molecular Biology, LSU Health Sciences Center, Shreveport, United States

**Competing interests:** The authors declare that no competing interests exist.

## Introduction

The integration of various aspects of the transcription of DNA into mature mRNA relies on combinatorial phosphorylations of the flexible scaffold structure formed by the RNA polymerase II (PolII) subunit Rpb1 C-terminal domain (CTD), which is comprised of repeats of the consensus heptad YSPTSPS (*Phatnani and Greenleaf, 2006*; *Hsin and Manley, 2012*). Genes transcribed by the PolII show a stereotypical pattern of CTD phosphorylation with phospho-S5 (S5P) peaking near the transcription start site (TSS) and phospho-S2 (S2P) accumulating towards the 3′-end of the transcribed region (*Buratowski, 2009*). Despite discrepancies, genome-wide analyses of CTD phosphorylation support that the CTD cycle is uniform across genes (*Kim et al., 2010*; *Mayer et al., 2010*; *Tietjen et al., 2010*; *Bataille et al., 2012*), and a broad body of evidence supports the seminal role of CTD S5P and S2P in transcriptional regulation and chromatin metabolism (*Buratowski, 2009*). However, recent works showed that contrary to S5P, S2P is dispensable in both fission yeast and budding yeast (*Cassart et al., 2012*; *Devos et al., 2015*) and only affects the steady-state level of a subset of mRNAs (*Coudreuse et al., 2010*; *Saberianfar et al., 2011*; *Sukegawa et al., 2011*). Therefore, a disconnect exists between the apparent uniform occupancy of the phosphorylated PolII and the gene-specific defects resulting from the disappearance of a phosphorylation; a similar case occurring with chromatin regulators (*Drogat and Hermand, 2012*; *Egloff et al., 2012*; *Weiner et al., 2012*).

The distribution of histone H3K4 and K36 methylation mirrors CTD phosphorylation due to the direct recruitment of the H3 methyltransferases Set1-COMPASS (for H3K4) and Set2 (for H3K36) by the S5P and S2P of PolII, respectively (*Ng et al., 2003*; *Keogh et al., 2005*). Set1 is the only H3K4

**eLife digest** The process of activating genes—known as gene expression—involves a number of steps. During the first step, the gene's DNA is copied or 'transcribed' to produce a molecule of messenger RNA. However, most of the DNA in a cell is wrapped around proteins called histones to make structures known as nucleosomes, and the DNA has to be unpacked to allow the enzymes that make messenger RNA to access it.

Cells regulate how the DNA is packed by attaching chemical groups to the histone proteins. Adding acetyl groups to histones usually causes the nucleosomes to unwrap and creates loosely packed DNA that helps with gene expression. On the other hand, the addition of methyl groups has the opposite effect.

RNA polymerase II is the enzyme that carries out transcription of messenger RNAs in all eukaryotic cells—that is, the cells of organisms like plants, animals, and fungi. Like all enzymes, RNA polymerase II is made of smaller building blocks called amino acids. One end of the RNA polymerase II enzyme, called the C-terminal domain (or CTD), contains a unique sequence of amino acids that serves as a scaffold to recruit other proteins involved in transcription and histone modifications. Different amino acids in this region of RNA polymerase II can be modified by the addition of phosphate groups. The pattern of these modifications is often thought of as a code and can influence which other proteins get recruited.

It remains poorly understood how RNA polymerase II regulates nucleosomes to allow transcription to occur. Materne, Anandhakumar et al. have now addressed this issue by engineering mutant yeast cells in which phosphate groups cannot be added to specific amino acids in the RNA polymerase II enzyme. Most genes were expressed as normal in these yeast cells, but a few hundred genes were not expressed.

Materne, Anandhakumar et al. then used a technique called MNase-Seq to map the position of nucleosomes across the genome and found that there were more nucleosomes near to start of these down-regulated genes. Further experiments showed that the addition of phosphate groups onto the RNA polymerase II is required to deplete the nucleosomes at the start of a gene called *ste11*, which allows transcription to occur.

Materne, Anandhakumar et al. also found that artificially tethering the enzyme that adds phosphate groups to the C-terminal domain to the start of the *ste11* gene was sufficient to oust nucleosomes and activate transcription by RNA polymerase II.

Future work will address if this newly discovered mechanism is implicated in the activation of specific patterns of gene expression during the development of more complex organisms.

methyltransferase in yeast, but it produces monomethylation, dimethylation, and trimethylation states. In budding yeast, H3K4me3 is strongest near the TSS, while H3K4me2 is highest just downstream, and H3K4me1 is dispersed throughout the length of the transcribed region (*Liu et al., 2005*; *Pokholok et al., 2005*). However, in vertebrates, the majority of H3K4me2 colocalizes with H3K4me3 in discrete regions nearby the promoter (*Ruthenburg et al., 2007*). Set2 targets H3K36me2 and me3 that are both highest near the 3′-end (*Krogan et al., 2003*; *Kizer et al., 2005*).

Despite a strong positive correlation between the H3K4 and K36 methylations and active PolII occupancy, their primarily function appears to be to repress histone acetylation and transcription because both serve as a binding platform for recruitment and the activation of histone deacetylase complexes (HDACs) including Set3 complex (SET3C) and Rpd3C(S) (*Kim and Buratowski, 2009*; *Buratowski and Kim, 2010*; *Govind et al., 2010*). This paradox may only be apparent because HDACs counteract unregulated initiation that could profit from the elevated nucleosome dynamics associated with acetylation during active transcription.

Known methyl-lysine-binding domains include the plant homeodomain (PHD) finger and the chromodomain. Available data support that these domains are responsible for the H3 methylation-dependent recruitment of HDACs. For example, the chromodomain protein Eaf3 is a subunit of Rpd3C(S) deacetylase and binds H3K36me (*Carrozza et al., 2005*; *Keogh et al., 2005*). Complementarily, the PHD finger protein Set3 is part of the SET3C complex and binds H3K4me2 to mediate deacetylation of histones in the 5′ regions (*Kim and Buratowski, 2009*; *Kim et al., 2012*).

Similarly, the PHD domain of the HDAC-associated ING2 protein mediates its binding to the dimethylated and trimethylated H3K4 at the promoters of proliferation genes (*Pena et al., 2006*; *Shi et al., 2006*).

How the balance between acetylation and deacetylation at promoters is regulated during transcription is poorly understood, and a role of the phosphorylated CTD in regulating that process is unknown. Here, we show that S2P affects promoter nucleosome occupancy at a subset of genomic loci. Mechanistic studies reveal that upon gene activation, the Sty1 MAP kinase directly phosphorylates the CTD S2 kinase Lsk1, which results in increased S2P nearby the promoter. We also show that the doubly phosphorylated S2P-S5P CTD has lower affinity for Set1 compared to the S5P CTD. Therefore, the peak of S2P at the promoter counteracts H3K4 methylation and the recruitment of histone deacetylases. Our results indicate that cellular signalling mediated by the PolII CTD directly controls promoter nucleosome dynamics and gene transcription of a subset of genes.

## Results

### CTD S2P regulates promoter nucleosome dynamics at a subset of genes

Authentic full-length *rpb1* CTD S2A (*S2A*) mutants in fission yeast and budding yeast are viable (*Coudreuse et al., 2010*; *Schwer and Shuman, 2011*; *Cassart et al., 2012*; *Drogat and Hermand, 2012*) and are barely affected for steady-state transcription or termination (*Lenstra et al., 2013*), suggesting that the function of S2P in the coupling between transcription and maturation is either not essential or ensured by redundant mechanisms. Yet, the transcription of a subset of genes is strongly defective in the fission yeast *S2A* mutant as typically shown for *ste11* that encodes the master regulator of sexual differentiation (*Otsubo and Yamamoto, 2012*; *Anandhakumar et al., 2013*), which results in sterility. In order to understand the molecular basis of this specific sensitivity of S2P, we performed a genome-wide screen for genetic interactions using synthetic genetic arrays (SGAs) with both the S2 kinase *lsk1* (*cdk12* that encodes the S2 kinase) deletion and the *S2A* mutant. Although both screens highlighted a genetic link between S2P and RNA maturation, about a third of the shared interactions found fell in chromatin remodelling and modifications (*Figure 1—figure supplement 1*, *Figure 1—source data 1*). This genetic connection between S2P and chromatin biology led us to perform a genome-wide mapping of nucleosome position and occupancy by MNase-Seq. We used the DANPOS bioinformatics pipeline to analyze the data, which defines three categories of nucleosome dynamics: position shift, fuzziness change, and occupancy change. These analyses revealed that 3.8% of the nucleosomes were dynamics in the S2A mutant compared to the wt (based on a false discovery rate <0.005), carrying mostly occupancy changes and position shifts (*Figure 1—source data 2*). Statistical analyses revealed that these dynamic nucleosomes are enriched in the promoter region encompassing −350 bp to +50 bp flanking the TSS (p-value < 2.2e-16, odds ratio = 2.82), and especially the −1 nucleosome (*Figure 1A*, left panel). A meta-gene analysis of the nucleosome occupancy signal for all protein-coding genes revealed higher occupancy around the TSS in the S2A mutant, and an 11 bp shift of the average −1 nucleosome toward the TSS (*Figure 1A*, middle panel). When selecting the 10% protein-coding genes whose promoter nucleosome-depleted region (NDR) shrinks the most in the absence of S2P (*Figure 1A*, right panel), the shift of the average −1 nucleosome rose to 44 bp (*Figure 1B*, left panel). In contrast, the 10% protein genes with the lower decrease in promoter NDR did not show a similar effect. In addition, the genes showing promoter dynamic nucleosomes in the absence of S2P tended to have larger NDRs and 5′-UTR (*Figure 1—source data 2*), two features reminiscent of the genes that we have previously characterized as showing a peak of S2P nearby the promoter during transcriptional activation (*Coudreuse et al., 2010*), including *ste11* (*Figure 1C*). In addition, we found that the 10% genes whose promoter NDR shrinks the most in the S2A mutant are highly enriched in this list of genes showing early S2P (Fisher's exact test p-value = 0.0013, odds ratio = 2.6) and in genes downregulated more than twofold in the S2A mutant (Fisher's exact test p-value = 0.0057, odds ratio = 1.4).

Importantly, even though a global nucleosome occupancy gain is also observed at the 3′ ends of protein-coding genes (TTS), it did not significantly affect the length of the NDR at TTS (*Figure 1D*).

Taken together, these analyses reveal an unexpected role of S2P in regulating nucleosomes occupancy and position at the promoter of 324 genes (*Figure 1—source data 2*).

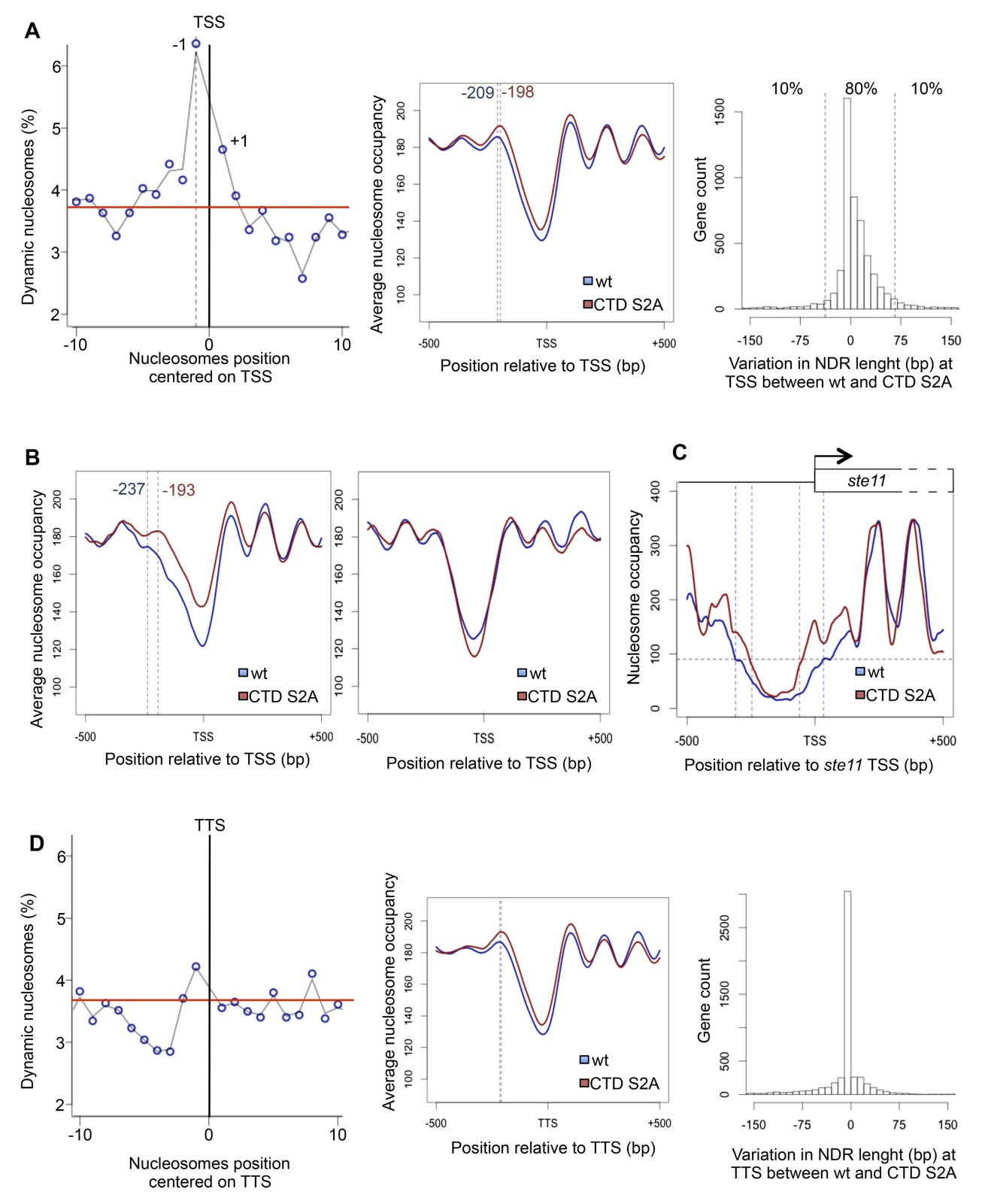

**Figure 1**. The RNA polymerase II S2P affects nucleosome dynamic at the promoter of a subset of genes. (**A**) Left panel: the percentage of dynamic nucleosomes aligned relative to the transcription start site (TSS) in the fission yeast genome. The red line indicates the average percentage of dynamic nucleosome over the genome (3.7%). Middle panel: meta-gene analysis of the nucleosome occupancy signal for all protein-coding genes near the TSS.

*Figure 1. continued on next page*

*Figure 1. Continued*

The distance between the TSS and the average −1 nucleosome midpoint position is indicated in blue for the wt and red for the S2A mutant. Right panel: distribution of the variation in nucleosome-depleted region (NDR) length at TSS between wt and the S2A mutant. The 10% genes showing the strongest increase or strongest decrease in NDR size are indicated. (**B**) Left panel: meta-gene analysis of the nucleosome occupancy signal for the 10% protein-coding genes showing the strongest decrease in NDR size at the TSS in the S2A mutant. The distance between the TSS and the average −1 nucleosome midpoint is indicated in blue for the wt and red for the S2A mutant. Right panel: meta-gene analysis of the nucleosome occupancy signal for the 10% protein-coding genes showing the strongest increase in NDR size at the TSS in the S2A mutant (blue: *wt*, red: *S2A*). (**C**) Nucleosomes occupancy nearby the promoter of *ste11* (blue: *wt*, red: *S2A*). (**D**) Left panel: the percentage of dynamic nucleosomes aligned relative to the TTS in the fission yeast genome. The red line indicates the average percentage of dynamic nucleosome over the genome (3.7%). Middle panel: meta-gene analysis of the nucleosome occupancy signal for all protein coding genes near the TTS. Right panel: distribution of the variation in NDR length at TTS between wt and the S2A mutant (blue: *wt*, red: *S2A*).

The following source data and figure supplement are available for figure 1:

**Source data 1**. List of the synthetic lethal genetic interactions uncovered with the *lsk1Δ* and *CTD S2A* mutant strains.

**Source data 2**. General features of the dynamic nucleosomes and the associated genes.

**Figure supplement 1**. Genome-wide synthetic lethal interaction mappings of the *lsk1Δ* and *rpb1 S2A* mutants link Rpb1 CTD S2P to chromatin biology.

## CTD S2P regulates histone occupancy and acetylation within the promoter region of *ste11*

Building on our previous work on *ste11*, we therefore analyzed the occupancy of H3 in the presence or absence of S2P, which revealed an increased level of H3 occurring specifically at the promoter of *ste11*, while the transcribed region was barely affected. The effect was not observed at the *adh1* locus whose transcription does not require S2P (*Figure 2A*). Nucleosome scanning provided a high-resolution picture of nucleosomes occupancy at the unusually large NDR of *ste11* with a marked increase in nucleosome abundance close to the +1 site in the *S2A* mutant (*Figure 2A*, *Figure 2—figure supplement 1A,B*). The loss of the NDR when S2P is abolished is a plausible explanation for the strong reduction of PolII occupancy and transcription observed previously (*Coudreuse et al., 2010*).

Importantly, the requirement of S2P for *ste11* transcription was independent of Set2, even though we could confirm that the molecular link between S2P and H3K36 methylation is conserved in fission yeast (*Figure 2—figure supplement 1C–E*).

In fission yeast, the transcription of the *ste11* gene is strongly induced by nitrogen or glucose starvation (*Coudreuse et al., 2010*), which is correlated with a specific increase in histone acetylation at the promoter (*Figure 2B*). In the absence of the S2 kinase Lsk1, no increase was observed (*Figure 2B*). Remarkably, the inhibition of Lsk1 using an analogue-sensitive mutant (Lsk1-as) led to an increase of the level of promoter H3 within 1 hr and abolished acetylation during *ste11* induction (*Figure 2C,D*). These data indicate that the phosphorylation of the CTD by Lsk1 dynamically regulates histone occupancy and acetylation over the *ste11* promoter.

Considering the well-established presence of S2P on the elongating PolII, the possibility exists that the increased nucleosome occupancy at the promoter would be a secondary effect resulting from an elongation defect in the S2A strain. To test this possibility, we analyzed the effect of the *leo1* mutant that encodes a component of the PAF complex that regulates elongation (*Mbogning et al., 2013*). The absence of *leo1* resulted in an increased occupancy of PolII over the *ste11* gene body (*Figure 2E*) and a decreased expression of *ste11* (*Figure 2F*). However, it did not affect the level of H3 at promoter (*Figure 2G*). The marked decrease of the level of the PolII over the entire *ste11* locus observed when S2P is absent (*Coudreuse et al., 2010*) and the increase occupancy of H3 at the *ste11* promoter (*Figure 2A*) are therefore not recapitulated in a mutant showing an elongation defect.

We next investigated the effect of Histone Acetyltransferase (HAT) and HDAC on the expression of *ste11* and their connection with S2P.

## The peak of S2P at the *ste11* promoter counteracts the S5P-Set1-SET3C pathway

We first tested if the decreased acetylation observed when S2P is abolished resulted from the loss of chromatin-associated Gcn5, the main HAT at work at the *ste11* locus (*Helmlinger et al., 2008*).

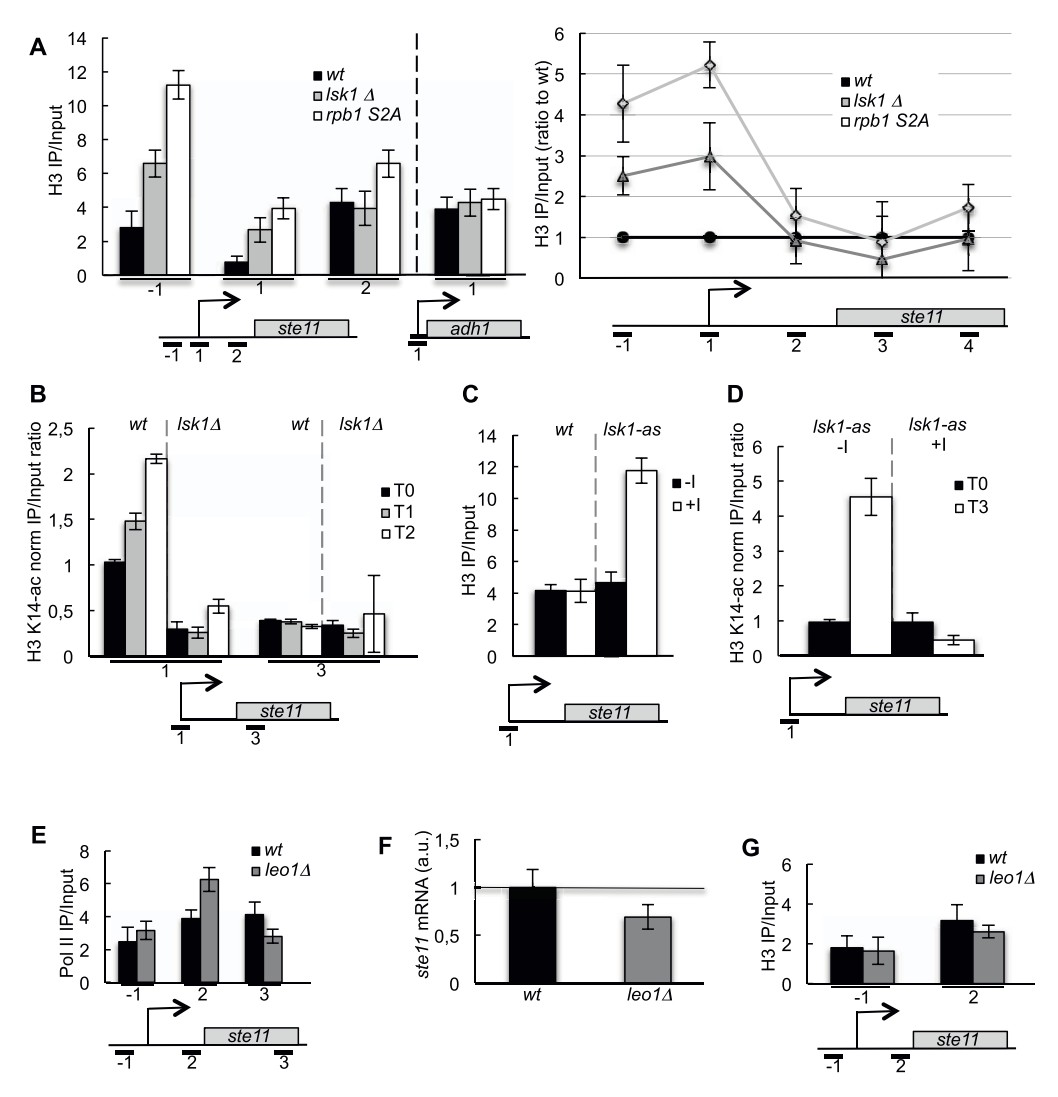

**Figure 2**. The CTD S2P regulates histone occupancy and acetylation at the promoter of the *ste11* gene. (**A**) Left panel: the occupancy of H3 measured by ChIP using the indicated amplicons in the 5′ proximal region of the *ste11* and *adh1* genes in the indicated strains. Right panel: similar to the left panel except that more amplicons were used and the data obtained in the *lsk1Δ* and *S2A* strains were normalized to the *wt* set as 1. (**B**) The *wt* and *lsk1Δ* strains were starved for nitrogen at the indicated time points (hours). The occupancy of acetylated H3 at the indicated places was determined by ChIP using anti-H3-K14-ac normalized against unmodified H3. (**C**) The occupancy of H3 was measured by ChIP using the indicated amplicon in a *lsk1-as* strain grown in the presence or absence of 10 μM 3-MB-PP1 for 1 hr. (**D**) A *lsk1-as* strain was cultured during vegetative growth (T0) and starved for nitrogen for 3 hr (T3). At T0, 10 μM 3-MB-PP1 (+I) or an identical volume of DMSO (−I) was added as indicated. The occupancy of acetylated H3 at the indicated places was determined by ChIP using anti-H3-K14-ac normalized against unmodified H3. (**E**) The occupancy of RNA polymerase II (8WG16 antibody) measured by ChIP using the indicated amplicons in the 5′ proximal region of the *ste11* gene in the indicated strains. (**F**) Relative quantification (RQ) of the *ste11* mRNA determined by quantitative Q-RT-PCR in the indicated strains. a.u.: arbitrary units. (**G**) The occupancy of H3 measured by ChIP using the indicated amplicons in the 5′ proximal region of the *ste11* gene in the indicated strains.

The following figure supplement is available for figure 2:

**Figure supplement 1**. Nucleosome scanning of the *ste11* 5′ region reveals increased occupancy in the CTD S2A mutant.

However, the level of Gcn5 was not decreased in the absence of S2P (*Figure 3—figure supplement 1A*), which led us to consider the alternative possibility that the S2P is required to counteract the effect of histone deacetylation by one or several histone deacetylase (HDAC). Supporting this possibility, a treatment with Trichostatin A led to an increase in the level of the *ste11* mRNA of sevenfold and 25 fold, respectively, in the *wt* and the mutants lacking S2P (*Figure 3—figure supplement 1B*). These results highlight the fact that chemically induced histone deacetylation bypasses the requirement of S2P for *ste11* transcription.

In order to test if the decreased histone acetylation observed the absence of S2P was a cause, or a consequence of the increased nucleosome occupancy over the NDR, we generated unacetylatable K14R (*H3K14R*) and K36R (*H3K36R*) histone H3 mutants (*Figure 3—figure supplement 1C*) and showed that they behaved similarly to the *S2A* strain, while a *H3K36Q*, which mimics constitutive acetylation had the opposite effect (*Figure 3—figure supplement 1D,E*). We conclude that the removal of a single acetylation site on H3 (either K14 or K36 was tested) is sufficient to cause an alteration of nucleosome occupancy within the promoter resulting in a defect in *ste11* transcription.

We next monitored the level of *ste11* mRNA in all the known fission yeast HDAC mutants (*Figure 3—figure supplement 1F*). The deletion of *hos2* led to a marked derepression of *ste11*, while other single mutants somehow affected *ste11* but never to the extend of the loss of *hos2*. The Hos2 enzyme is part of the conserved SET3C deacetylase complex that is recruited in budding yeast by S5P either directly or through the methylation of H3 lysine 4 (H3K4) by Set1, which itself depends on the S5P (*Ng et al., 2003*; *Kim and Buratowski, 2009*; *Govind et al., 2010*). ChIP experiments revealed that Hos2 peaks at the *ste11* promoter and that its recruitment was strongly dependent upon the S5-phosphorylated CTD as its occupancy was decreased in a CTD S5A mutant, which is kept viable by fusing the human capping enzyme Mce1 (*Schwer and Shuman, 2011*) (*Figure 3C*). Contrary, the absence of *lsk1* resulted in an increased occupancy of the Hos2 HDAC at the *ste11* promoter (*Figure 3D*), which likely explains the marked decrease in H3 acetylation observed when Lsk1 is absent. Consistent with the antagonistic role of S5P and S2P in the chromatin association of Hos2, abolition of S5P suppressed the defect in *ste11* expression observed in the *S2A* mutant and restored wild-type level of the *ste11* mRNA (*Figure 3E*). Most likely, the suppression is mediated through the loss of the Hos2 HDAC as the deletion of *hos2* in the *S2A* mutant behaved similarly to the *S5A* mutant (*Figure 3F*).

Taken together, these genetic interactions suggest that S2P counteracts the S5P-Hos2 (SET3C) pathway to increase histone acetylation at the promoter of *ste11*. As indicated above, Set1 is recruited by the CTD S5P, and the resulting H3K4 methylation is a prerequisite for the binding of the SET3C HDAC. Accordingly, the deletion of *set1* resulted in a derepression of *ste11*, (*Figure 4A*). It was recently shown that Set1 can repress transcription independently of the methylation of its target, H3K4 (*Lorenz et al., 2014*). However, we also observed a twofold increase of the level of the *ste11* mRNA in a strain harbouring a H3K4R mutant that cannot be methylated, suggesting that the effect of Set1 at the *ste11* locus is mediated mainly through H3K4 methylation (*Figure 4A*). Importantly, the deletion of *set1* suppressed the sterility of the *lsk1* mutant (*Figure 4B*).

Based on these data, we speculated that the burst of S2P at the promoter could interfere with the binding of Set1 to the CTD S5P. In order to test this possibility, we expressed plasmid born NLS-GST-CTD (full-length) fusions harbouring wt, S5A, or S2A repeats in fission yeast. Preliminary experiments revealed that the wild-type version was phosphorylated on both S2 and S5, while the S2A or S5A versions lacked the corresponding S2P or S5P, respectively (*Figure 4C*). When expressed in a Set1-TAP strain, only the S2A version (S5 phosphorylated) could be immunoprecipitated by Set1-TAP. No interaction was observed with the S5A CTD version, confirming the dependency of Set1 to S5P to bind the CTD. Importantly, the doubly S2P-S5P phosphorylated CTD did not bind Set1-TAP, indicating that the presence of S2P interferes with the binding of Set1 to S5P (*Figure 4C*).

Taken together, the previous data indicate that the timely burst of S2P in the promoter region of *ste11* counteracts the S5P-dependent methylation and deacetylation occurring at the promoter by physically displacing Set1. This model raises the question of the recruitment of Lsk1 in the 5′ region of the transcribed unit.

## MAP kinase-dependent phosphorylation of the S2 kinase Lsk1 regulates its specific recruitment in the 5′ region of the *ste11* gene

Mass spectrometry analyses of TAP-purified Lsk1 complexes together with proteome-wide analyses (*Wilson-Grady et al., 2008*; *Beltrao et al., 2009*) revealed that some of the MAP kinase consensus

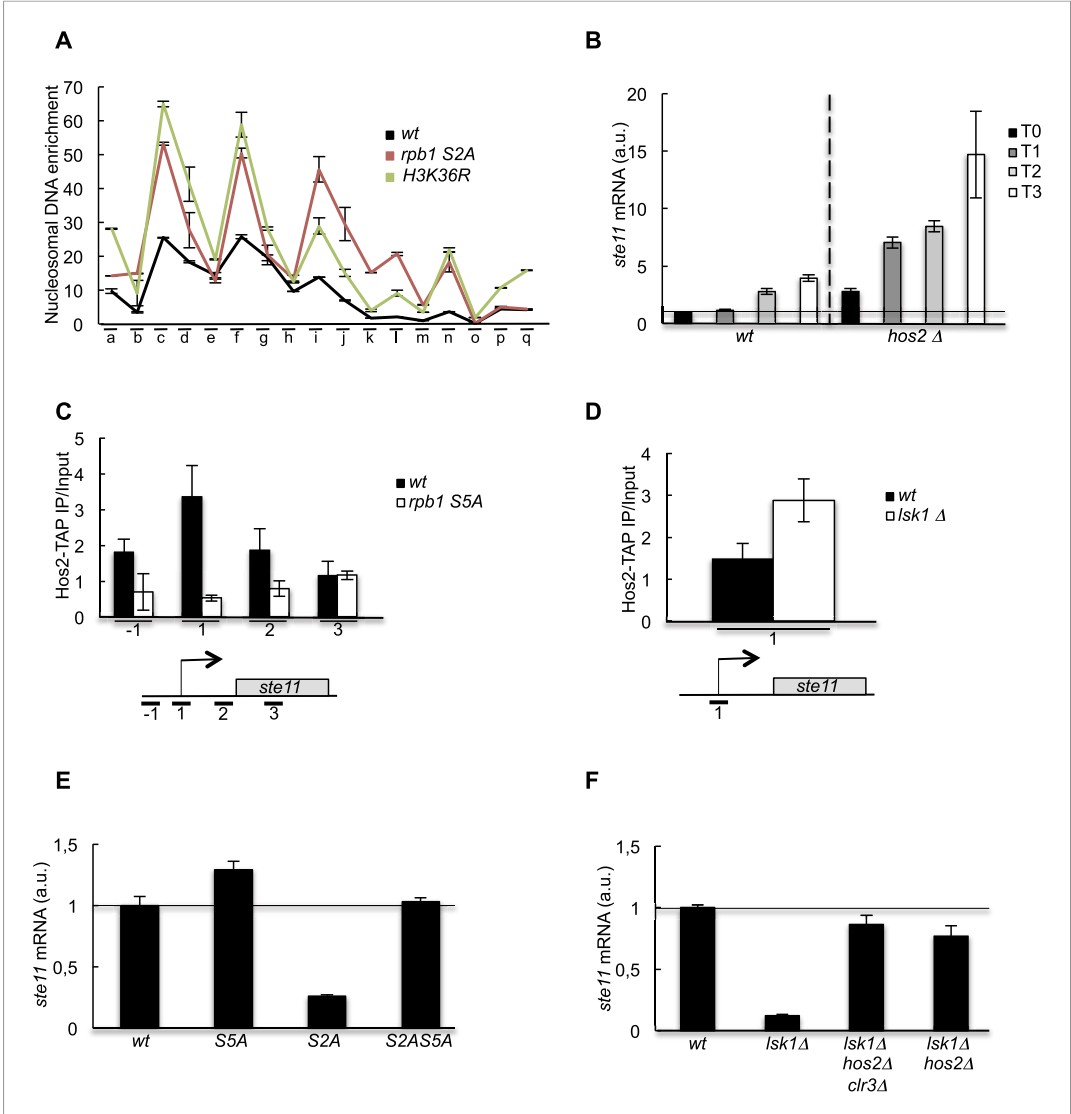

**Figure 3.** The increase of CTD S2P nearby the promoter region is necessary to reverse the CTD S5P-dependent deacetylation of nucleosomes. (**A**) Nucleosome scanning analysis of the *S2A* and *H3K36R* strains. Nucleosomal DNA enrichment at the indicated positions of the *ste11* locus was determined by ChIP experiment on MNase-digested chromatin. Data are presented as the average of two independent experiments along with the SEM. (**B**) RQ of the *ste11* mRNA determined by Q-RT-PCR in the *wt* and *hos2Δ* strains during nitrogen starvation at the indicated time points (hour). a.u.: arbitrary units. (**C**) The occupancy of Hos2-TAP was measured by ChIP using the indicated amplicons at the *ste11* locus in a *wt* strain and a *CTD S5A* mutant. Each column represents the averaged value ± SEM (n = 4). (**D**) Identical to **C** with a *wt* strain and an *lsk1Δ* mutant. (**E–F**) RQ of the *ste11* mRNA determined by quantitative Q-RT-PCR in the indicated strains. a.u.: arbitrary units.

The following figure supplements are available for figure 3:

**Figure supplement 1**. HDAC-dependent control of *ste11* expression by CTD S2P independently of Gcn5.

**Figure supplement 2**. Untagged control experiments for the chromatin immunoprecipitations.

sites (PXS/TP) found within the N-terminal region of Lsk1were phosphorylated in vivo. All the phosphorylated residues are located within a long N-terminal extension that is present in all the Cdk12 orthologs but with low sequence conservation compared to the canonical C-terminal CDK domain (*Figure 5—figure supplement 1A,B*). An N-terminal truncated version of Lsk1 rescues the

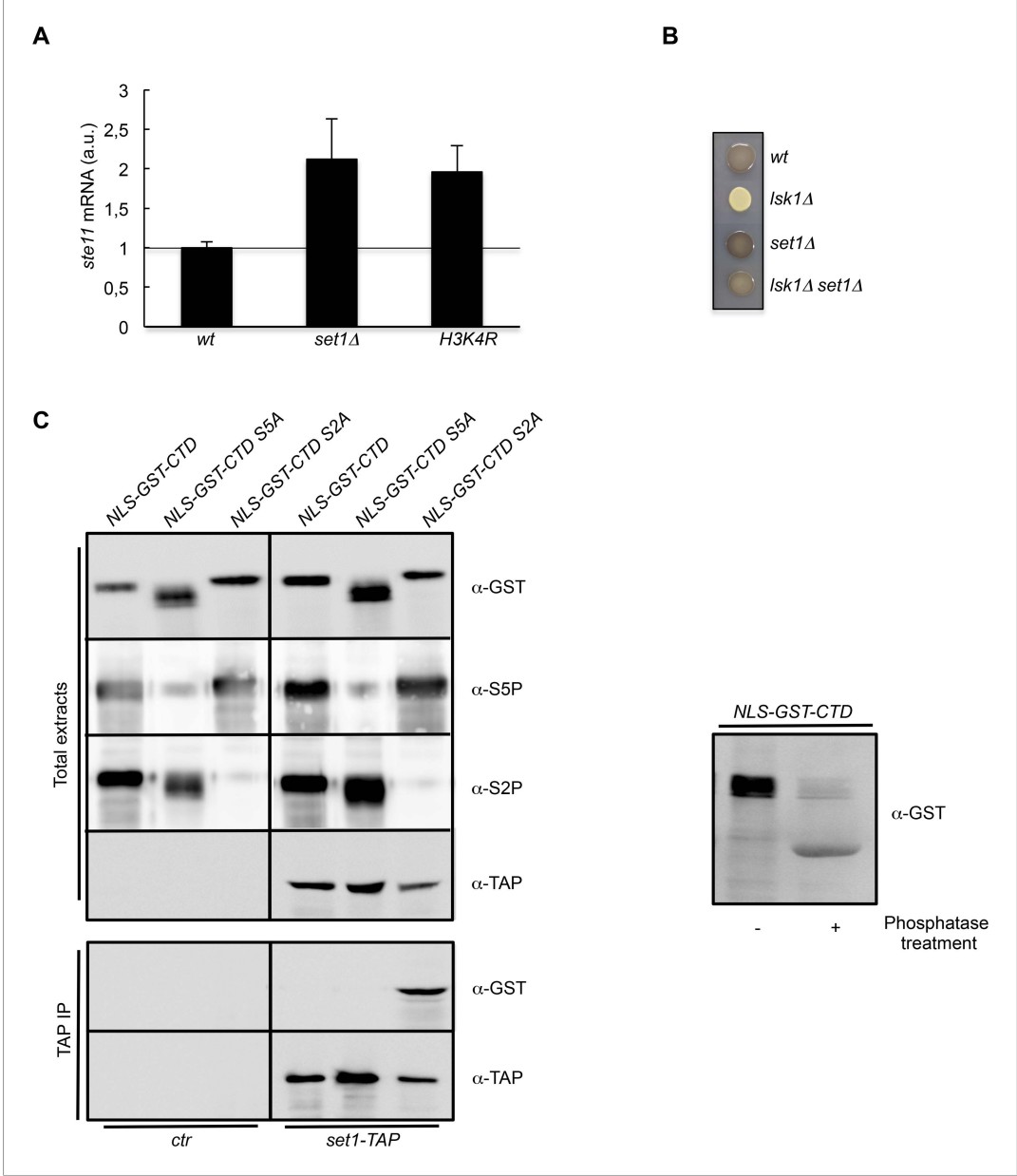

**Figure 4**. The S2P antagonizes Set1 binding to the CTD. (**A**) RQ of the *ste11* mRNA determined by quantitative RT-PCR using the ΔΔct method in the *wt, set1Δ*, and *H3K4R* strains. Each column represents the averaged value ± SEM (n = 2). a.u.: arbitrary units. Note that the wild-type control is different in the case of the *set1Δ* and *H3K4R* strains because the histone mutant is constructed in a background where only one copy of the histone H3 is retained (*Figure 3—figure supplement 1C*). Isogenic strains are used as control and set as 1. (**B**) Heterothallic wild-type, *lsk1Δ, set1Δ*, and *set1Δ lsk1Δ* strains were plated for 48 hr on mating medium before iodine staining to reveal sterility. (**C**) NLS-GST-CTD fusions containing wt, S2A, or S5A repeats were expressed in fission yeast from a pREP-3 plasmid. Protein extracts from the *ctr* and *set1-TAP* strains expressing these fusions were probed by Western blot using the anti-GST, anti-S5P, anti-S2P, or anti-TAP antibodies as indicated (Top panel-Total extracts). After TAP immunoprecipitation, the resulting beads were probed by Western blot using anti-GST or anti-TAP antibodies (Bottom panel-TAP IP). Right panel: a protein extract from the *ctr* strain expressing the NLS-GST-CTD fusion was treated or not by a phosphatase and probed by Western blot using anti-GST antibodies.

thermosensitive defect of *lsk1* deletion and shows in vitro kinase activity similar to the *wt*, which suggests a regulatory rather than catalytic role for the N-terminal extension (*Figure 5—figure supplement 1C,D*). It was previously shown that the Sty1 MAP kinase (*Sanso et al., 2011*) could

phosphorylate Lsk1 in vitro (*Sukegawa et al., 2011*). We confirmed and expanded these data and showed that 12 sites including loosely consensus sequences needed to be eliminated in order to abrogate the phosphorylation (*Figure 5A*). The deletion of the N-terminal region of Lsk1 (Lsk1 ΔN) or the mutation of the 12 sites phosphorylated by Sty1 (Lsk1 12SA) severely impaired the induction of the *ste11* mRNA in a way reminiscent of the *sty1* deletion (*Figure 5B*). Contrary, the mutation of the four canonical MAPK sites to glutamic acid, mimicking constitutive phosphorylation, increased the level of *ste11* mRNA in both vegetative growth and differentiation suggesting that the Sty1-dependent signal transduction through the Lsk1 CTD kinase is a rate limiting step in *ste11* transcriptional regulation.

ChIP experiments revealed that the abolition of MAPK signalling in the Lsk1 ΔN or the Lsk1 12SA mutants resulted in a marked decrease in the occupancy of Lsk1 and S2P along the *ste11* 5′ regulatory regions. A very similar reduction of Lsk1 occupancy was observed in the absence of Sty1 (*Figure 5C*). When the phosphorylation of Lsk1 by Sty1 was abolished, a marked decrease in the occupancy of PolII was also observed (*Figure 5D*).

Taken together, these data suggest a model where that the Sty1 MAPK pathway dynamically regulates the specific recruitment of Lsk1 in the 5′ region of the *ste11* gene, which is required for its transcription. In order to test this model, we designed a chimeric Lsk1 kinase where the regulatory N-terminal extension was replaced by the DNA-binding domain (High Mobility Group [HMG]) of the Ste11 transcription factor (*Figure 6A*). Considering that Ste11 binds its own promoter, we anticipated that the DNA-binding domain should bring Lsk1 at the *ste11* promoter independently of the nutritional status of the cells. The HA-tagged chimeric HMG-Lsk1 kinase was properly expressed from the thiamine repressed pREP vector (*Figure 6B*). Remarkably, the induction of the HMG-Lsk1-HA protein led to a more that sixfold increase in the level of the *ste11* mRNA, while the removal of thiamine had a modest effect (*Figure 6C*). ChIP experiments indicated that the HMG-Lsk1-HA kinase was properly recruited at the *ste11* promoter, which resulted in an increase in S2P (*Figure 6D,E*). The effect was not observed at the *adh1* promoter. The results show that the forced recruitment of Lsk1 at the *ste11* promoter is sufficient to induce transcription, independently of a nitrogen or glucose starvation.

## Several HDAC complexes are targeted by CTD S2P during gene activation

Many important insights into transcriptional regulation have come from studies on glucose repression of the *Spinv1/ScSUC2* gene that encodes an invertase required to use alternative carbon sources (*Celenza and Carlson, 1986*; *Hoffman and Winston, 1991*; *Ahn et al., 2012*). Remarkably, in a way very reminiscent of *ste11*, the derepression of *inv1* in low glucose also required S2P, the MAP kinase consensus sites in Lsk1 N-terminal region and the Sty1 kinase (*Figure 7A*). Moreover, as shown for *ste11*, the transcriptional induction of *inv1* was concomitant to an early increase of S2P within the promoter (*Figure 7B*). In order to determine the relative increase of S2P over the *ste11* and *inv1* loci after 1 hr of induction, we first normalized the level of S2P (based on the 3E10 antibody) on the total level of PolII (based on the 8WG16 antibody). We next determined the ratio of normalized S2P observed at T1 on T0. This revealed that PolII reached its highest level of S2P close to the promoter during the induction of *ste11* and *inv1*. By contrast, the increase of S2P over the *fbp1* locus (a gene induced in the same conditions as *inv1* (*Hirota et al., 2008*) was strongly biased to the 3′ end of the gene (*Figure 7—figure supplement 1*), reminiscent of the 'canonical' occupancy profile of S2P, and *fbp1* induction was totally independent of S2P (data not shown).

Within the *inv1* regulatory region, the absence of S2P resulted in decreased acetylation and increased occupancy of histones, which was associated with a marked decrease of PolII occupancy (*Figure 7C–E*). Both the S5A and the *set1* deletion mutants fully suppressed the defect resulting from the lack of S2P (*Figure 7F*). However, the deletion of Hos2 (SET3C) did not affect *inv1* expression (data not shown). A previous genome-wide analysis reported that glucose repression of *inv1* requires the Rpd3C(L) (Clr6)-dependent promoter histone deacetylation (*Wiren et al., 2005*). The SET3C and Rpd3C(L) complexes both harbour a binder for H3K4me: the Set3 PHD finger and the Png2 ING domain proteins, respectively (*Shi et al., 2006*; *Nicolas et al., 2007*; *Shi et al., 2007*). We therefore tested if the deletion of *png2* could suppress the defect of *inv1* expression in the absence of S2P, which was the case (*Figure 7F*).

These data indicate that, similarly to the case of *ste11*, the phosphorylation of S2 within the CTD plays a critical role to reverse the methylation-dependent recruitment of the Rpd3C(L) complex

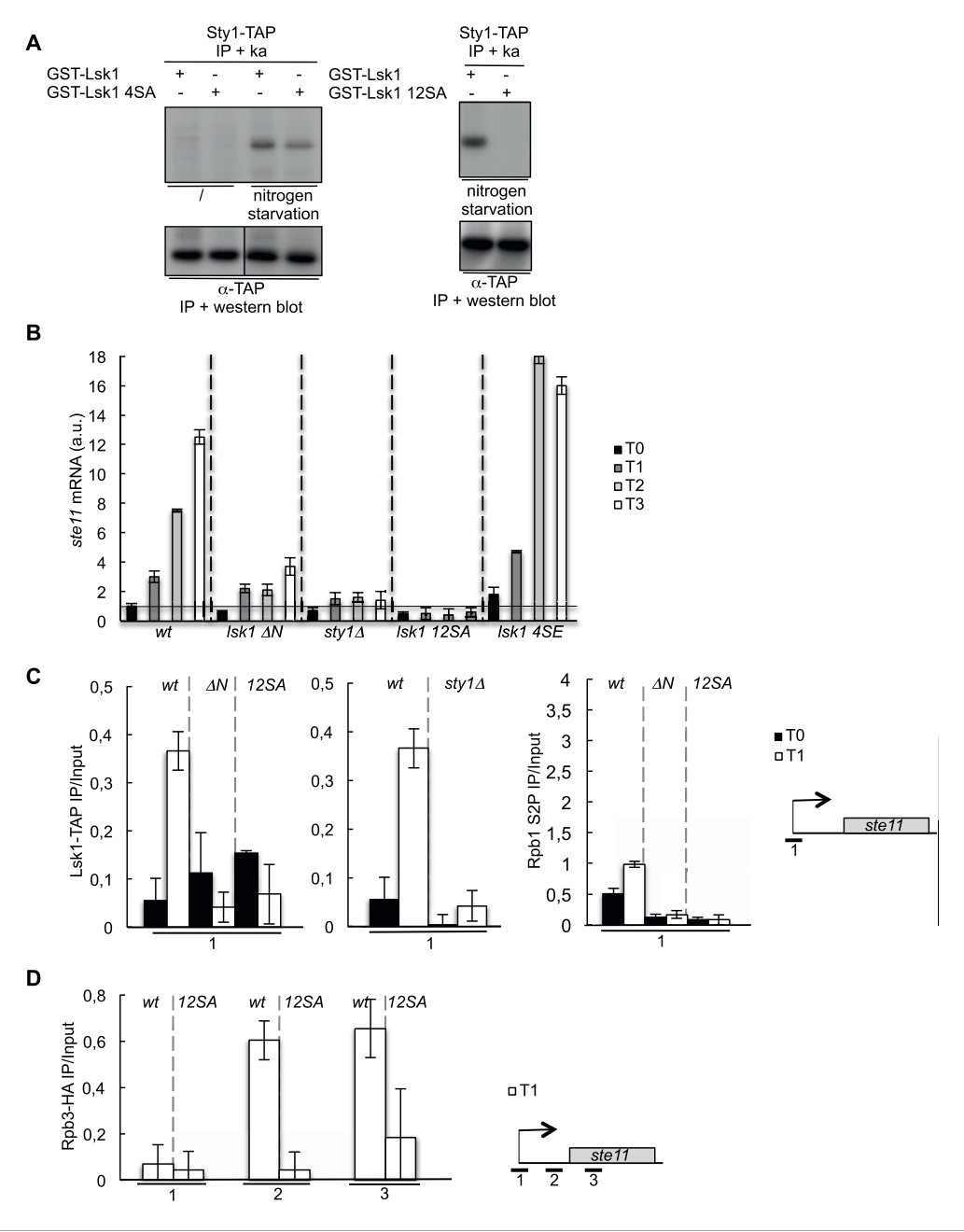

**Figure 5**. The phosphorylation of Lsk1 by Sty1 is required for CTD S2 phosphorylation in the 5′ proximal region of the *ste11* gene. (**A**) The Sty1-TAP protein was precipitated and used on beads for in vitro kinase assays using *wt* and mutated (4SA or 12SA) forms of the GST-Lsk1 protein as indicated. The MAP kinase was activated by nitrogen starvation. The amount of precipitated proteins was estimated by Western blot analysis using peroxidase–antiperoxidase (PAP). (**B**) RQ of the *ste11* mRNA determined by Q-RT-PCR in the indicated strains during nitrogen starvation at the indicated time points (hour). a.u.: arbitrary units. *lsk1 4SE* refers to a mutant where the four residues fitting the perfect MAPK consensus were mutated to glutamic acid to mimic phosphorylation. (**C**) Left panel, the occupancy of Lsk1-TAP and the Lsk1ΔN-TAP and Lsk112SA-TAP mutants was measured by ChIP using the indicated amplicons. Middle panel, same experiment in *wt* or *sty1Δ* strains. Right panel, the occupancy of S2P was measured by ChIP using the indicated amplicons in the indicated strains. These experiments were performed during vegetative growth (T0) and early nitrogen starvation (T1). (**D**) The occupancy of RNA Pol II was measured by ChIP in the indicated strains using the indicated amplicons. The experiment was performed during early nitrogen starvation (T1).

*Figure 5. continued on next page*

*Figure 5. Continued*

The following figure supplement is available for figure 5:

**Figure supplement 1**. The phosphorylation of the CTD S2 kinase Lsk1 on 12 sites in vivo is not required for its activity.

---

to the promoter of *inv1* and allows transcriptional activation upon glucose starvation and MAP kinase signalling.

## Discussion

The phosphorylation of the PolII CTD and the methylation of H3K4 are probably the best markers of active transcription, but it is unclear how their interplay is regulated to affect gene expression. Here, we show that cellular signalling through the CTD code directly modulates H3K4 methylation-dependent deacetylation of promoter nucleosomes at some specific loci, which is required for gene activation during cell differentiation and metabolic adaptation.

Genome-wide MNase-Seq in a strain where S2P is absent revealed that a subset of 324 genes displays specific changes of promoter nucleosome dynamic, defined as position shift, fuzziness change, and occupancy change as defined by the DANPOS bioinformatics pipeline, which leads to a shorter NDR at TSS. These target genes tend to have larger than average NDR and 5′-UTR, which may indicate that they are under more complex regulation. This is clearly the case of *ste11* and *inv1*, two model genes that we have analyzed in more details. These analyses revealed that the phosphorylation of S2 is critical for gene activation by reversing the effects of histone deacetylation by HDAC at the promoter.

Histone deacetylases including Rpd3C(L) and the SET3C complexes localize to promoter regions through their capacity to bind H3K4me, which links them to Set1 and S5P. In the subset of genes we have analyzed, we propose that during transcriptional induction, the local increase of S2P nearby the promoter prevents the efficient association of Set1 with the S5P CTD (*Figure 8*). Interestingly, previous work revealed a spread of the Set1-dependent H3K4me2 and H3K4me3 marks 3′ into the bodies of genes in the absence of Ctk1, the CTD S2 kinase in budding yeast (*Wood et al., 2007*; *Xiao et al., 2007*). These data were interpreted to mean that S2P phosphorylation acts as a barrier to prevent the CTD S5P-dependent association of Set1, which explain the restricted pattern of H3K4 dimethylation and trimethylation classically observed. We propose that the temporary, MAP kinase dependent, targeting of S2P at the promoter of target genes acts similarly. In this context, it was also shown that the presence of S2P upstream of S5P within a peptide abolishes the recognition by an anti-S5P antibody (*Hintermair et al., 2012*), again suggesting that S2P somehow interferes with the recognition of the neighbour S5P, which we observe using an NLS-GST-CTD fusion expressed in vivo (*Figure 4*).

The decrease in histone methylation and deacetylation following the rise of S2P leads to a burst in acetylation and transcription initiation (*Figure 8*). Despite the fact that the SET3C and the Rpd3C(L) complexes have very different subunits composition, they are both responding to the signalling through CTD S2P because they harbour the histone methyl-lysine-binding domain proteins Set3 and Png2, respectively. A remaining question is why different HDAC complexes are required in the case of *ste11* and *inv1*. The Rpd3C(L) has been proposed to regulate transcriptional burst frequency, while the SET3C complex is thought to modulate the burst size (*Hnisz et al., 2012*; *Weinberger et al., 2012*), so the involvement of different HDAC complexes is likely related to gene-specific induction dynamics.

In budding yeast, the current model of H3K4 methylation-dependent histone deacetylation proposes that Set1, recruited by the CTD S5P, deposits the H3K4me2 mark, which in turn recruits the SET3C via the Set3 PHD finger, localizing the Hos2 and Hst1 HDAC subunits of SET3C in 5′ transcribed regions (*Kim and Buratowski, 2009*; *Buratowski and Kim, 2010*). It was also reported that SET3C is directly recruited by the CTD S5P, whereas it is the interaction with methylated H3 that is required for the deacetylation activity (*Govind et al., 2010*). Whatever the exact mechanism, the promoter occupancy of the HDACs complexes requires S5P. Considering the evolutionary conservation of all the proteins implicated in this model, we envision that the process may be

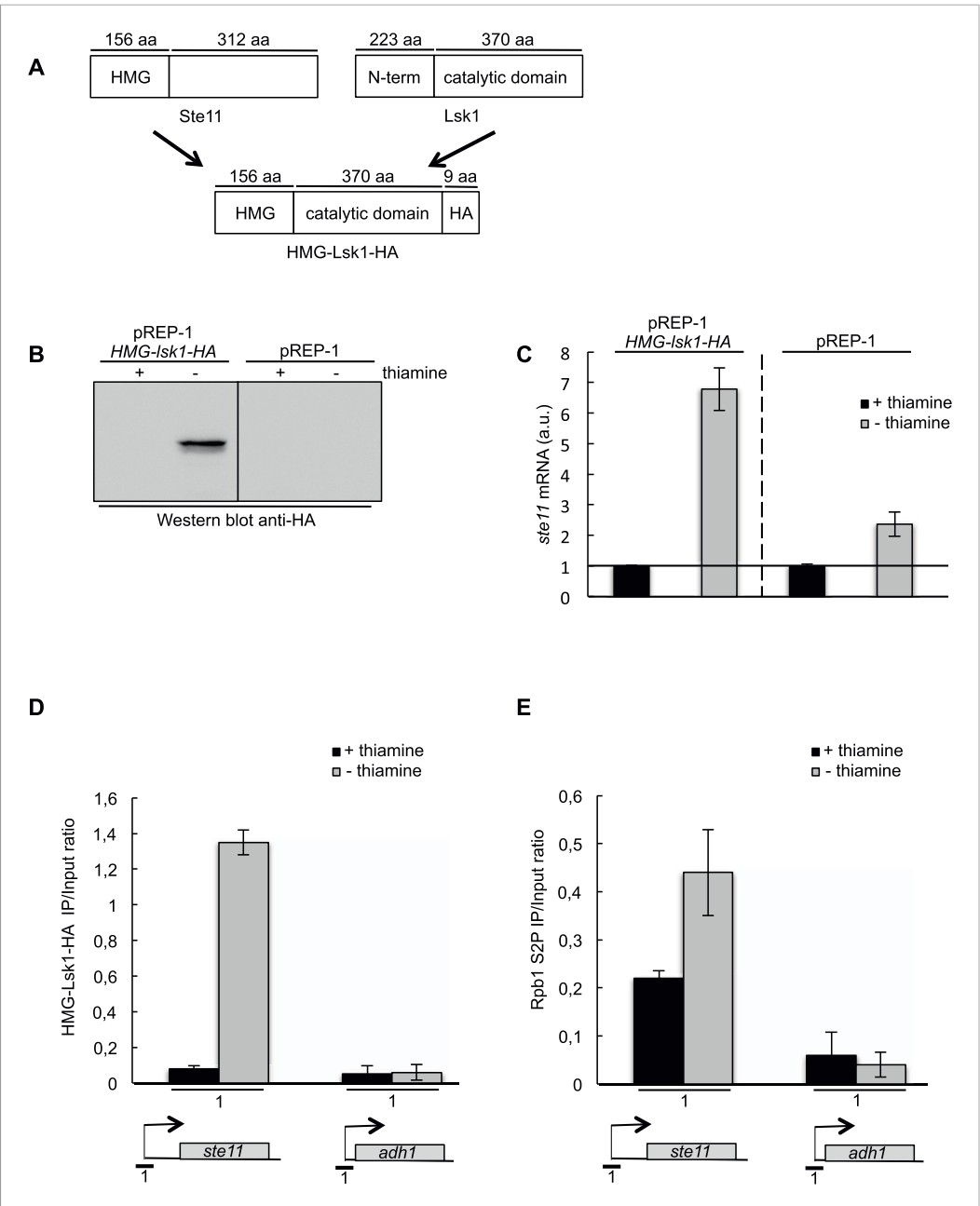

**Figure 6**. A chimeric HMG-Lsk1-HA protein induces *ste11* expression independently of nitrogen starvation. (**A**) A schematic of the chimeric HMG-Lsk1-HA protein. The wild-type Ste11 and Lsk1 proteins are depicted and the size of various regions indicated in amino acids. (**B**) Western blot analysis (anti-HA) of a wild-type strain containing the pREP-1 *HMG-lsk1-HA* plasmid or the corresponding empty vector and grown for 22 hr in the presence or absence of thiamine (that represses expression) as indicated. (**C**) RQ of the *ste11* mRNA determined by quantitative RT-PCR using the ΔΔct method in a wild-type strain containing the indicated plasmids and grown for 22 hr in the presence or absence of thiamine as indicated. a.u.: arbitrary units. (**D**) The occupancy of HMG-Lsk1-HA at the *ste11* and the *adh1* promoters was measured by ChIP (anti-HA) using the indicated amplicon in a wild-type strain containing the pREP-1 *HMG-lsk1-HA* plasmid and grown for 22 hr in the presence or absence of thiamine as indicated. Each column represents the mean percentage immunoprecipitation value ± SEM (n = 2). (**E**) Similar to **D** except that the immunoprecipitation was performed with an anti-S2P antibody.

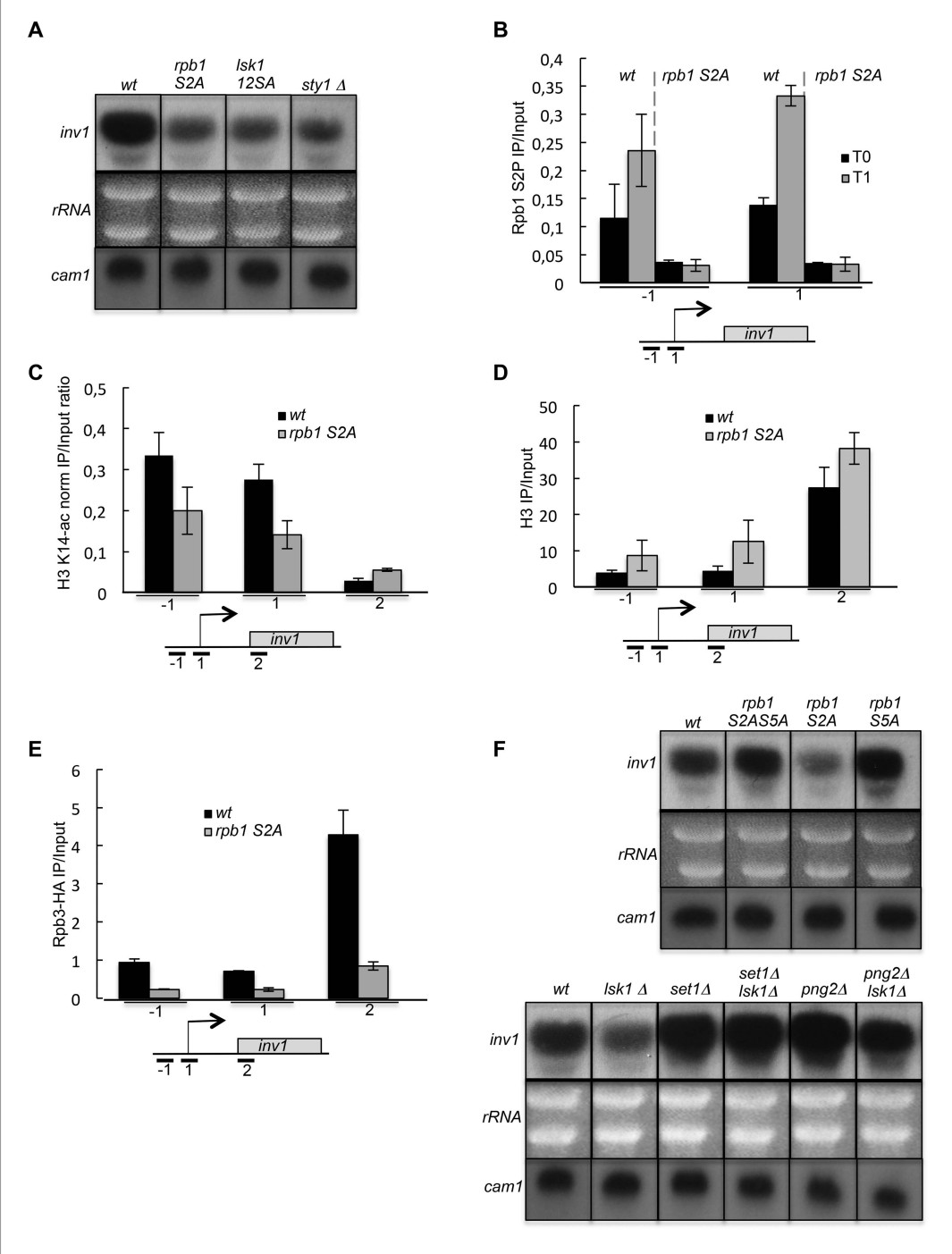

**Figure 7.** The induction of *inv1* upon glucose deprivation is regulated by S2P-dependent control of promoter nucleosome acetylation. (**A**) Northern blot analysis of *inv1* expression in the indicated strains after an hour shift from high glucose (2%) to low glucose (0.1%). Note that the mRNA is not detected in high glucose (not shown). Ethidium bromide strained ribosomal RNAs and the level of expression of the *cam1* gene are shown as loading controls. (**B**) The occupancy of S2P was measured by ChIP using the indicated amplicons in the 5' proximal region of the *inv1* gene in the indicated strains. These experiments were performed in high-glucose medium (T0) and after 1 hr in low-glucose medium (T1). (**C**) The *wt* and *S2A* strains were cultured in low glucose for 1 hr to induce *inv1*. The occupancy of acetylated H3 was determined by ChIP using anti-H3-K14-ac normalized against unmodified H3 (**D**). The location of the amplicons used in Q-PCR is indicated. (**E**) The occupancy of RNA Polymerase II was measured by ChIP in the indicated strains using the indicated amplicons in the 5' proximal region of the *inv1* gene. The strains were cultured

*Figure 7. Continued*

in low glucose for 1 hr to induce *inv1*. (**F**) Northern blot analyses of *inv1* expression in the indicated strains after an hour shift from high glucose (2%) to low glucose (0.1%). Ethidium bromide strained ribosomal RNAs and the level of expression of the *cam1* gene are shown as loading controls.

The following figure supplement is available for figure 7:

**Figure supplement 1**. The RNA polymerase II CTD S2P reaches its maximal level early during the induction of the *ste11* and *inv1* genes.

operating at some target genes in other eukaryotes. In support to this view, the budding yeast S2 kinase, Ctk1, is also phosphorylated within the N-terminal region (*Chi et al., 2007*; *Breitkreutz et al., 2010*). The reported defect of the *ctk1* mutant for gametogenesis (*Lee and Greenleaf, 1991*) led us to test the requirement of Ctk1 for the increased acetylation of promoter histones during the induction of the master regulator of meiosis, *IME1*, in diploid budding yeast cells. The absence of *ctk1* abrogated the burst of promoter histones acetylation (*Figure 8—figure supplement 1A*) and resulted in low expression (*Figure 8—figure supplement 1B*). Remarkably, the defect observed in the absence of CTD S2P was suppressed by the deletion of the Set3 subunit of the SET3C HDAC complex (*Figure 8—figure supplement 1B*), in agreement with the data obtained in fission yeast. These data, together with the recent discovery of a gene-specific role of Cdk12 in the control of gene expression in human cells (*Blazek et al., 2011*), support the conservation of the molecular mechanism we report here, which opens further dissection in higher eukaryotes. Interestingly, in *Drosophila*, the UpSet protein, an ortholog of yeast Set3, also recruits Rpd3-containing HDACs to developmental genes and its PHD domain was proposed to recognize methylation on H3K4, establishing a dependency to Set1 (*Rincon-Arano et al., 2012*, *2013*). The most common phenotype associated with the deletion of UpSet occurs during gametogenesis, similarly to yeast.

Our work also shows that S2 phosphorylation within the CTD responds to cellular signalling. Upon activation, the Sty1 MAP kinase phosphorylates the CTD S2 kinase Lsk1 on multiple sites spread over the long N-terminal extension of the kinase. We show that the phosphorylation of Lsk1 is required for the 5′ peak of CTD S2P during the induction of the *ste11* and *inv1* genes, while it does not appear to be required for the kinase activity itself. This dependency on MAP kinase phosphorylation likely explains why only a subset of promoters is regulated in this manner. Considering that MAP kinases, including Sty1, are recruited to their target genes, they are likely to confer this specificity. Remarkably, the artificial tethering of the S2 kinase Lsk1 at the *ste11* promoter is sufficient to induce *ste11* expression, independently of nitrogen starvation and MAP kinase phosphorylation. Further studies are required to understand how the phosphorylation of the conserved N-terminal extension of the S2 kinase modulates its recruitment. Previous work revealed that MAPK signalling exerts a multilayered control on gene expression (*de Nadal et al., 2011*). Our data demonstrate that MAP kinase also directly controls CTD phosphorylation, which expands the emerging theme that chromatin-associated proteins are key responders to environmental cues (*Weiner et al., 2012*).

## Materials and methods

### General methods

Fission yeast growth, gene targeting, and mating were performed as described (*Bamps et al., 2004*; *Fersht et al., 2007*). TAP purification was performed as described (*Guiguen et al., 2007*). TAP immunoprecipitation and kinase assay on GST-CTD were previously described (*Drogat et al., 2012*). Inhibitors (1-Nm-PP1 and 3-Mb-PP1) of the analogue-sensitive mutant kinases were purchased from Toronto Research Chemicals (Toronto, Canada). Trichostatin A was purchased from Millipore (Billerica, MA) and used at 50 µg/ml. GST-fusion proteins were expressed and purified using the GE kit according to the manufacturer instructions with the following variations: growth was performed at 18°C and induction performed at 0.5 mg/ml IPTG. The expression of *ste11* was induced by nitrogen starvation or by the addition of methionine, as described (*Schweingruber et al., 1998*; *Coudreuse et al., 2010*). The expression of *inv1* was induced by shifting cells grown in YE 2% glucose medium

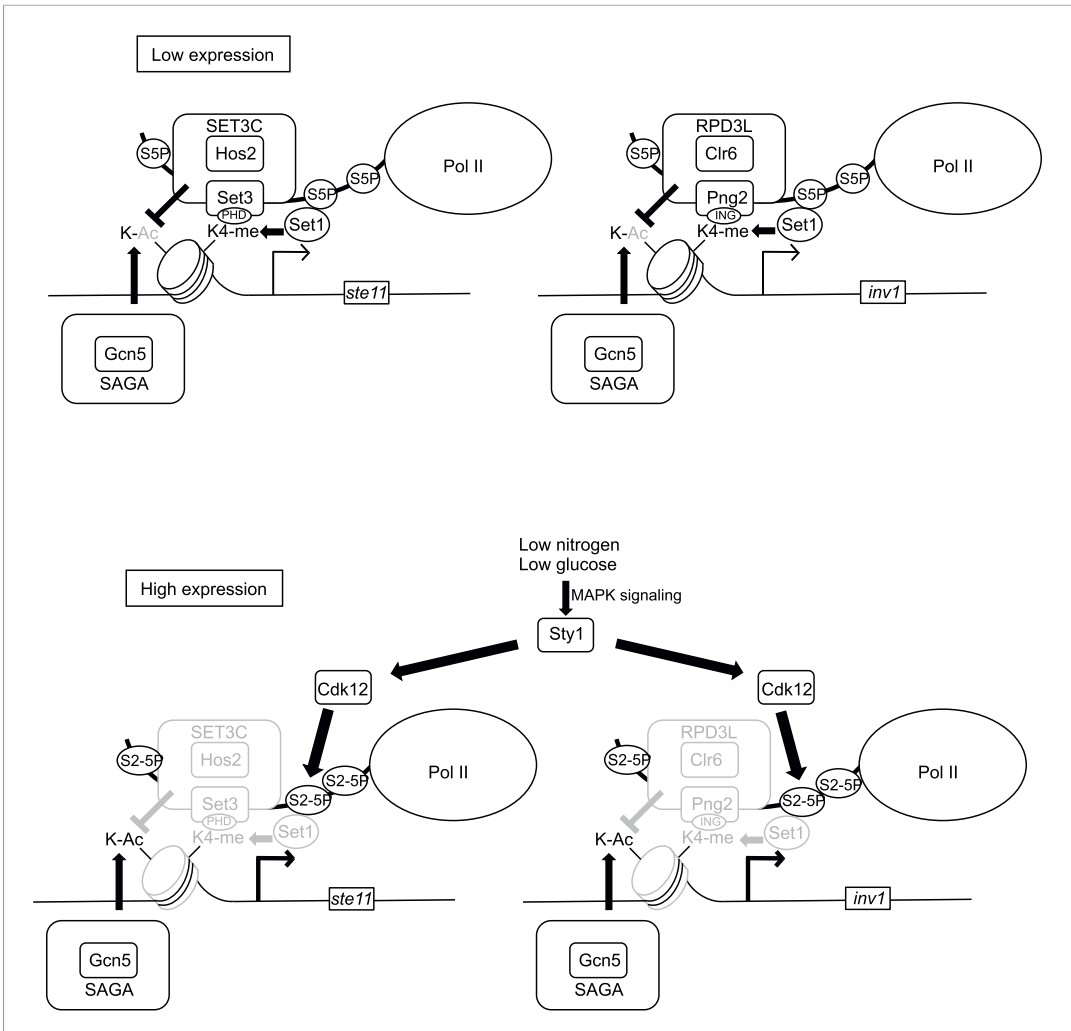

**Figure 8**. Model of the control of promoter nucleosome dynamics by CTD S2 phosphorylation. When gene expression is low in non-induced conditions, the phosphorylation of the CTD on S5 recruits the Set1 methylase and the SET3C or the RPD3L HDAC complexes. The Set3 and Png2 subunits of these two complexes mediate interactions with the methylated (di or tri-) H3K4. Upon induction of gene expression, the Sty1 MAP kinase is activated and phosphorylates the CTD S2 kinase Lsk1, which is required for an increase in S2P nearby the promoter. We propose that the S2P-S5P CTD displaces the HDACs complexes, which leads to a local increase in nucleosome acetylation and dynamics. In the absence of CTD S2P, the nucleosomes occupancy over the NDR is high, which impedes the RNA pol II access and efficient transcription.

The following figure supplement is available for figure 8:

**Figure supplement 1**. The role of CTD S2P in promoter histone acetylation is conserved in budding yeast.

to YE 0.1% glucose/glycerol 3% for 1 hr. Western blot were performed with anti-GST (Sigma), peroxidase–antiperoxidase (PAP) (Sigma), anti-S2P (3E10, Covance), anti-S5P (H15, Covance) antibodies.

Phosphatase (λ-phosphatase, New England Biolabs) treatment was performed as described. Iodine staining was performed as described (*Bauer et al., 2012*).

## ChIP and quantitative RT-PCR
Chromatin immunoprecipitations were performed using a Bioruptor (Diagenode, Belgium) and Dynabeads (Invitrogen, Calrsbad, CA). Note that when a tagged protein is used (Lsk1-TAP, Rpb3-HA, Gcn5-myc, Hos2-TAP), the IP/T ChIP ratio is shown in the main figures and the corresponding

untagged control strain treated simultaneously is shown in *Figure 3—figure supplement 2*. Total RNA was prepared and purified on Qiagen RNeasy. Quantitative RT-PCR was performed using the ABI high capacity RNA-to-cDNA. The untreated sample was used as a reference and the *act1* mRNA was used for normalization. Antibodies used in ChIP were PAP (Sigma), anti-Rpb1 S2P (Chromotek), anti-RNA polymerase II (Millipore), anti-HA (Covance), anti-H3 (abcam), anti-H3K14ac (Millipore), anti-H3K36ac (Millipore), anti-H3 K4Me2 (Millipore), anti-H3K4Me3 (Millipore), anti-H3K36me (a kind gift of Jean-Christophe Andrau), anti-Myc (Covance).

For all ChIP experiments, each column represents the mean percentage immunoprecipitation value ± SEM (n = 2–4). All oligonucleotides used are listed in *Supplementary file 1*.

## Quickchange mutagenesis, plasmid construction

All the site-directed mutagenesis were performed using the Quickchange kit (Stratagene) following the instructions of the manufacturer.

To generate the pREP-1 *HMG-lsk1-HA*, the sequence of the insert was designed by gene synthesis (Integrated DNA Technologies) and the resulting fragment was cloned in the pREP-1 plasmid using the BamHI-SmaI restriction sites.

## Cloning and expression of NLS-GST-CTD

In order to clone the full-length versions of NLS-GST-CTD, the wt, S2A, and S5A GST-CTD inserts were amplified from pGEX4-T1 vectors (*Coudreuse et al., 2010*) adding the coding sequence for the following NLS PKKKRKVA and the XhoI-BglII restriction sites. These inserts were cloned in pREP-3 in SalI-BamHI, and the resulting plamsids transformed in fission yeast.

## Integration of mutants

The integration of the histone mutants is described in *Figure 3—figure supplement 1C*. The *lsk1* mutants were integrated at the *lsk1* locus by replacement and 5-FOA selection in a *lsk1::ura4* strain. All the mutants were amplified from a plasmid after Quickchange mutagenesis using the Expand TAQ polymerase (Roche).

## SGAs

Due to the sterility of the *lsk1::natR* deletion strain and the *rpb1 S2A-natR* strain, both strains were first transformed with a plasmid expressing the *ste11* gene, which rescues the sterility. To test for genetic interaction, the query strains were mated to the array of single deletion strains (Bioneer deletion library) on SPA plates. All steps were performed manually using V&P Scientific pin replicators allowing either 96 or 384 wells format. Immediately following the transfer of cells onto the mating plate. Cells were then allowed to sporulate at 26°C for 3 days, and mating plates were subsequently transferred to 42°C for 3 days in order to eliminate unmated/unsporulated cells, thereby enriching for spores. Following heat treatment, spores were transferred on YES plates and allowed to germinate for 2 days at 32°C. To select for recombinant double-mutant haploids, cells were arrayed on YES plates containing G418 and nourseothricin and allowed to grow for a further 2 days.

## Nucleosome scanning

A culture of 500 ml of fission yeast cells was grown to OD 0.5 at 32°C and crosslinked with 7 ml of formaldehyde 37% for 20 min at 25°C, 60 rpm. The crosslink was stopped by the addition of 27 ml of Glycine and cells were pelleted. The pellet was resuspended in preincubation solution (Citric acid 20 mM, Na2HPO4 20 mM, Ethylenediaminetetraacetic acid [EDTA] pH 8 40 mM) supplemented with 100 μl β mercaptoethanol/50 ml and incubated 10 min at 30°C. The cells were centrifugated and resuspended in 10 ml of Sorbitol 1 M/Tris pH 7.4 50 mM buffer containing 200 μl Zymolase (0.01 g/ 200 μl water) and incubated 20 min at 30°C (40 min when Edinburgh Minimal Media [EMM] medium was used). After centrifugation, the pellet was resuspended in 7.5 ml NP buffer (Sorbitol 1 M, NaCl 50 mM, Tris pH 7.4, 10 mM, MgCl$_2$ 5 mM, CaCl$_2$ 1 mM, NP-40 0.75%) supplemented with 7.6 ml NP buffer + 0.5 μl β mercaptoethanol + 400 μl spermidine 10 mM) and split into 2 Falcon tubes (Total and MNase treated). Add 50 μl MNase (32 units) to one tube and incubate 20 min at 37°C without agitation. Add 500 μl Stop buffer, 200 μl RNase A (0.4 mg/ml), and 225 μl proteinase K (20 mg/ml) and incubate at 65°C overnight. Potassium acetate was added (1.25 ml of a 3 M solution) and the

mix was incubated 5 min on ice. After phenol extraction, 200 µl NaCl 5 M, 1.7 µl Glycogene (20 mg/ml), and 3.5 ml of isopropanol were added. After precipitation and ethanol wash, the pellet was resuspended in 200 µl of TE buffer. The samples were run on agarose gel (1.5%) and the bands corresponding to the mononucleosomes were cut and purified with Qiagen. Q-PCR with the primer pairs of the set of overlapping amplicons was performed. Nucleosomal DNA enrichment calculated as the ratio between the amounts of PCR product obtained from DNA samples generated from the mononucleosomal gel purification to that of the input (total) DNA.

### TAP immunoprecipitation and kinase assays
These assays were performed as described (*Guiguen et al., 2007*).

### Northern blot and Q-RT-PCR
Total RNA was prepared as described (*Guiguen et al., 2007*) and purified on Qiagen RNeasy. Total RNA (15–30 µg) was separated on gel and transferred on nitrocellulose. Hybridization of a multiprimed labelled probe covering the *inv1* open reading frame was performed overnight at 42°C. Q-RT-PCR was performed using the ABI high capacity RNA-to-cDNA following the instructions of the manufacturer. The untreated sample was used as a reference and the *act1* mRNA was used for normalization. In all Q-RT-PCR experiments, each column represents the averaged value ± SEM (n = 2).

Note that while studying the induction of *inv1*, Northern blots were preferred to Q-RT-PCR because the *act1* mRNA showed variations during the course of the experiment.

### MNase-seq
The preparation of mononucleosomal DNA and the sequencing of mononucleosomal DNA were previously described in details (*Soriano et al., 2013*). The nucleosome sequencing data have been deposited in the GEO database under the accession number GSE59768.

### MNase-seq data processing and detection of dynamic nucleosomes and NDR analyses
MNase-seq data were processed and dynamic changes were detected using DANPOS (Dynamic Analysis of Nucleosome Positioning and Occupancy by Sequencing—https://code.google.com/p/danpos/). Clonal reads (determined by their very high coverage compared to the mean coverage across the genome based on a Poisson p-value cut-off) were removed from the reads previously mapped on the *Schizosaccharomyces pombe* genome with BWA (http://bio-bwa.sourceforge.net). Variation in size resulting from MNase treatment was compensated by shifting each read toward the 3′ direction for half of the estimated fragment size. Nucleosome occupancy was then calculated as the quantile-normalized count of adjusted reads covering each base pair in the genome. After this processing, DANPOS calculates the differential signal at single nucleotide position based on a Poisson test. Dynamic nucleosomes are then identified by peak calling on these signals. TSS and TTS-associated NDR length were quantified as the length of the longest DNA segment whose proximal border is located closer than 65 bp from the TSS (or TES) with nucleosome occupancy levels lower than an arbitrary threshold (mean(occupancy$_{genome}$) − standard_deviation(occupancy$_{genome}$)) at any point.

The processed data can be visualized on the web browser http://nucleosome.usal.es/fgb2/gbrowse/S2A/.

## Acknowledgements
We thank the GEMO laboratory for discussions and the ADRE team for excellent support. We thank Beate Schwer, Michael Buck, Karl Ekwall, Songtao Jia, Jason Tanny, and Robin Allshire for reagents, strains, and help with large-scale data sets. We thank Antonin Morillon, Mathieu Rougemaille, and Yota Murakami for critical reading of the manuscript. This work was supported by grants BFU2011-28804 and Consolider Ingenio CSD2007-00015 from the Spanish Ministerio de Economía y Competitividad to FA, and grants FRFC 2.4510.10, Credit aux chercheurs 1.5.013.09, MIS F.4523.11, Ceruna and Marie Curie Action to DH. DH is a FNRS Research Associate. This work is dedicated to the memory of Jean Vandenhaute.

# Additional information

## Funding

| Funder | Grant reference | Author |
|---|---|---|
| Ministerio de Economía y Competitividad | BFU2011-28804 / CSD2007-00015 | Francisco Antequera |
| Fonds De La Recherche Scientifique - FNRS | FRFC 2.4510.10 | Damien Hermand |
| Fonds De La Recherche Scientifique - FNRS | CR 1.5.013.09 | Damien Hermand |
| Fonds De La Recherche Scientifique - FNRS | MIS F.4523.11 | Damien Hermand |

The funders had no role in study design, data collection and interpretation, or the decision to submit the work for publication.

## Author contributions

PM, Conception and design, Acquisition of data, Analysis and interpretation of data; JA, VM, IS, CM, LQ, Acquisition of data, Analysis and interpretation of data; CY-S, DH, Conception and design, Analysis and interpretation of data, Drafting or revising the article; EH, Conception and design, Analysis and interpretation of data, Contributed unpublished essential data or reagents; FA, Conception and design, Analysis and interpretation of data

# Additional files

## Supplementary file

• Supplementary file 1. Oligonucleotides used in this study.

## Major dataset

The following dataset was generated:

| Author(s) | Year | Dataset title | Dataset ID and/or URL | Database, license, and accessibility information |
|---|---|---|---|---|
| Materne et al, | 2015 | Promoter nucleosome dynamics regulated by signaling through the CTD code | http://www.ncbi.nlm.nih.gov/geo/query/acc.cgi?acc=GSE+59768 | Publicly available at the NCBI Gene Expression Omnibus (Accession no: GSE59768). |

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
