## [Decision Letter]

[Editors’ note: a previous version of this study was rejected after peer review, but the authors submitted for reconsideration. The previous decision letter after peer review is shown below.]

Thank you for choosing to send your work entitled “Promoter nucleosome dynamics regulated by signaling through the CTD code” for consideration at *eLife*. Your full submission has been evaluated by James Manley (Senior editor), a Reviewing editor, and three peer reviewers, and the decision was reached after discussions between the reviewers. Based on our discussions and the individual reviews below, we regret to inform you that your work cannot be considered further for publication in *eLife*.

Although we recognize the potential novelty of the work describing a function of Ser-2 phosphorylation at the promoter, we believe that the phenomenon is likely not general, but most importantly in the absence of mechanism it only remains as an observation.

Reviewer #1:

The authors identified chromatin remodellers and chromatin modifiers in a synthetic screen analyzing mutants of Pol II CTD serine 2 (S2A) and the CTD S2 kinase lsk in *S. pombe*. Further analysis revealed that nucleosome dynamics and occupancy at promoters was altered in CTD S2A mutants in a subset of genes, including the genes *ste11* and *inv1*. Molecular and genetic experiments identified altered histone acetylation as a possible reason for increased histone occupancy at promoters.

Histone deacetylation at promoters is under the control of H3K4 methylation and the H3K4 methylation dependent recruitment of HDACs SET3C and RPD3 to promoters. The authors show that CTD S2 phosphorylation can block the recruitment of HDACs to promoters and thereby contribute to gene activation. Further, phosphorylation of kinase Lsk and its recruitment to promoters is growth regulated and under the control of MAP kinases. Mutation of MAP kinase specific phosphorylation sites in the N-terminal domain of Lsk prevents its recruitment to promoters, phosphorylation of CTD S2, and thereby the dissociation of HDACs from promoters. In line with these observations the authors can activate *ste11* expression after artificial tethering of LSK 1 at the *ste11* promoter. While the association of HDACs with promoters is CTD S5-P dependent the authors suggest that the combined phosphorylation of S5 and S2 can abolish this association, leading to hyperacetylation of chromatin and gene activation.

The assumption the CTD S2-P dependent mechanism of gene activation described here for *S. pombe* could be a more general mechanism is supported by supplementary data in paper. Inactivation of Ctk1, the *S. cerevisiae* CTD S2 kinase, abrogates promoter histone acetylation and results in low gene expression.

All together this is a very detailed and carefully performed analysis and all conclusions are supported by data.

How CTD S2-P marks mechanistically can abrogate the association of HDACs with promoters remains open but this question is beyond this paper.

Reviewer #2:

The manuscript by Hermand and coworkers investigates a functional link between CTD phosphorylation at serine-2 and histone dynamics in the yeast *Schizosaccharomyces pombe*. The authors provide a large number of experimental data ranging from genome-wide synthetic genetics arrays (SGAs) for growth phenotypes to locus-specific chromatin modifications. However, the presented results fail to provide compelling evidence for their thesis that 'promoter nucleosome dynamics is regulated by the CTD code'. Most importantly, the provided data do not discriminate between cause and effect or even bystander effects.

Many results and conclusions of this study have multiple interpretation issues.

First, the authors present the combined results from two genome-wide SGAs using a CTD-S2 kinase deletion and an *rpb1 CTD-S2A* mutant allele. As expected the identified genes display a significant overlap, but it is troublesome that only single subunits of chromatin regulatory complexes were identified. Also, there is little evidence of a systematic enrichment of ATP-dependent chromatin remodellers, which regulate nucleosome dynamics. Surprisingly, the authors did not investigate the identified Rsc1 and Swr1 genetic interactions in any detail. Next, the nucleosome density is determined in the rpb1 *CTD-S2A*, but not in the *lsk1* deletion. Compared to wild type yeast the authors observe small differences in global MNase sensitivity and they make conclusions on nucleosome dynamics. This is highly surprising. Nucleosome dynamics is defined as nucleosome turnover or half-lives at specific genomic loci. Therefore, the conclusion that 'these analyses reveal an unexpected role of S2P in regulation nucleosome dynamic' is flawed.

Next, the authors focus on the histone modifications at the *ste11* and *inv1* genes for locus-specific effects. The rationale for this is sketchy and not linked to the results from the SGA screens as the H3K4 and H3K36 (de)methylation or H3K14 (de)acetylation machineries were not identified in these screens. Also, the reason to analyze involvement of the PAF complex (via the LEO1 gene) in CTD-S2 phosphorylation dependent regulation of *ste11* is not clear. The conclusion that 'defects in initiation or elongation cannot explain the increased level of histone H3 observed at the *ste11* promoter in the absence of S2P' is not substantiated, as the authors did not analyze RNA polymerase II occupancy at *ste11* in the CTD-S2A strain.

Reviewer #3:

This paper follows up earlier results showing that a CTD serine 2 to alanine mutant has gene-specific, rather than general, effects in *S. pombe*. A synthetic lethal screen found many interactions between the S2A mutant or Lsk1/Cdk12 (S2 kinase) and chromatin-related factors. Nucleosome mapping revealed a subset of genes with altered nucleosome positions near promoters in these mutants. The rest of the paper focuses on one of these genes (*ste11*), with a few other experiments done on the *inv1* gene. The story then gets a little confusing. The model proposed is that these promoters are repressed by H3K4 methylation recruiting either SET3C or Rpd3L HDACs. To activate the genes, the MAPK Sty1 phosphorylates Lsk1, which in turn phosphorylates S2 near the *ste11* promoter to block H3K4 methylation and subsequent HDAC recruitment. It's a complicated model, and the many experiments needed to look at each of these steps makes the paper difficult to follow.

Major issues:

1) A big gap in the paper is understanding why only a small number of promoters are affected, since CTD phosphorylation and H3K4 methylation are seen at all active genes. At several points in the paper, the text suggests this is a general mechanism, but the data don't support that. What specifies which promoters are repressed, and is Sty1 recruited to these promoters specifically? The change in the NDR suggests remodelers may be involved—maybe these promoters are more dependent on SWI/SNF? Are there overlapping ncRNAs causing methylation over the promoter regions?

2) A critical element of the model is that double S2P+S5P no longer binds Set1 complex (and/or HDACs). This should be tested physically, maybe by peptide binding.

[Editors’ note: further revisions were requested before acceptance.]

Thank you for choosing to send your work entitled “Promoter nucleosome dynamics regulated by signaling through the CTD code” for consideration at *eLife*. Your submission has been considered by James Manley and the Reviewing Editor who oversaw the initial review.

We remain unconvinced of the general significance of the observations and are still concerned regarding the limited mechanistic insight. However, if you can: 1) show that the phenomena observed (ser-2 phosphorylation) occurs at a significant number of promoters in yeast; and 2) provide more solid insights into the mechanism, then we would be willing to consider a new manuscript on this topic.

---

## [Author Response]

Thank you very much for reviewing our manuscript entitled “Promoter nucleosome dynamics regulated by signaling through the CTD code”. I am aware of the fact that this manuscript is somehow challenging the current view of role of CTD S2P during transcription, which is precisely why we submitted it to *eLife*.

Reviewer #2:

*The manuscript by Hermand and coworkers investigates a functional link between CTD phosphorylation at serine-2 and histone dynamics in the yeast* Schizosaccharomyces pombe*. The authors provide a large number of experimental data ranging from genome-wide synthetic genetics arrays (SGAs) for growth phenotypes to locus-specific chromatin modifications. However, the presented results fail to provide compelling evidence for their thesis that 'promoter nucleosome dynamics is regulated by the CTD code'. Most importantly, the provided data do not discriminate between cause and effect or even bystander effects*.

This is a comment with which I strongly disagree. The core of our conclusion is twofold:

a) The S2 kinase Lsk1 (Cdk12) is regulated by MAP kinase dependent phosphorylation, which influences its chromatin distribution;

b) CTD S2P directly affects the occupancy of promoter nucleosomes, which leads to transcription activation.

There are quite some descriptive experiments in the paper that are necessary to give a detailed picture of the involvement of CTD S2P in gene activation. But in addition, we also provide experiments where we have interfered with the experimental system in various ways to demonstrate the causality between promoter CTD S2P and the activation of transcription. For example, our first conclusion is tested by mutating the phosphorylation sites of Lsk1 to a phosphomimmick or non phosphorylatable residues, which consistently uncouples the regulation of the S2 kinase from the MAP kinase (Figure 5). Beside several genetic evidences that CTD S2P is targeted to the promoters of some genes (*ste11* and *inv1*), we also engineered a chimeric S2 kinase that we can artificially tether to the *ste11* promoter, which results in transcriptional activation and bypasses the MAP kinase regulation (Figure 6). Contrary to what referee 2 states, these experiments clearly discriminate between a cause and an effect.

*Many results and conclusions of this study have multiple interpretation issues*.

*First, the authors present the combined results from two genome-wide SGAs using a CTD-S2 kinase deletion and an* rpb1 CTD-S2A *mutant allele. As expected the identified genes display a significant overlap, but it is troublesome that only single subunits of chromatin regulatory complexes were identified. Also, there is little evidence of a systematic enrichment of ATP-dependent chromatin remodellers, which regulate nucleosome dynamics*.

This is rather expected for a deletion library: some of the gene deletions are missing from the library, essential or slowly growing. Moreover, this is a minor experiment in the paper that could be removed without affecting the conclusions.

*Surprisingly, the authors did not investigate the identified Rsc1 and Swr1 genetic interactions in any detail*.

We found 66 genetic interactions. I should add that we have a second manuscript in preparation identifying the RSC complex as the ultimate target of the pathway we describe. Indeed, the CTD S2P dependent burst of acetylation we have described leads to the recruitment of RSC that affects nucleosome occupancy at the *ste11* promoter. We believe that this is beyond the scope of the current paper, which is already packed with data.

*Next, the nucleosome density is determined in the* rpb1 CTD-S2A*, but not in the* lsk1 *deletion*.

The referee himself indicated in his previous sentence about the SGA screen that the S2A and the *lsk1* deletion are expected to give a significant overlap. We would be happy to show the data for the *lsk1* deletion.

*Compared to wild type yeast the authors observe small differences in global MNase sensitivity and they make conclusions on nucleosome dynamics. This is highly surprising. Nucleosome dynamics is defined as nucleosome turnover or half-lives at specific genomic loci. Therefore, the conclusion that 'these analyses reveal an unexpected role of S2P in regulation nucleosome dynamic' is flawed*.

*Next, the authors focus on the histone modifications at the* ste11 *and* inv1 *genes for locus-specific effects. The rationale for this is sketchy and not linked to the results from the SGA screens as the H3K4 and H3K36 (de)methylation or H3K14 (de)acetylation machineries were not identified in these screens. Also, the reason to analyze involvement of the PAF complex (via the LEO1 gene) in CTD-S2 phosphorylation dependent regulation of* ste11 *is not clear*.

The SGA screens look for synthetic lethality. Why would the referee expect us to find H3K4 and K36 demethylases or HDAC? Based on our model, their deletion should rather result in synthetic improvement and rescue (and indeed the deletion of the Hos2 HDAC rescues the deletion of lsk1 (Figure 3).

*The conclusion that 'defects in initiation or elongation cannot explain the increased level of histone H3 observed at the* ste11 *promoter in the absence of S2P' is not substantiated, as the authors did not analyze RNA polymerase II occupancy at* ste11 *in the CTD-S2A strain*.

This exact experiment was shown in our previous study (Coudreuse et al., Current Biology 20, 2010, Figure 3) as referred in the paper. This is a minor aspect of the paper that could be removed without affecting the conclusions.

Reviewer #3:

*Major issues*:

1) A big gap in the paper is understanding why only a small number of promoters are affected, since CTD phosphorylation and H3K4 methylation are seen at all active genes. At several points in the paper, the text suggests this is a general mechanism, but the data don't support that. What specifies which promoters are repressed, and is Sty1 recruited to these promoters specifically? The change in the NDR suggests remodelers may be involved—maybe these promoters are more dependent on SWI/SNF? Are there overlapping ncRNAs causing methylation over the promoter regions?

We really don’t suggest that the mechanism we describe is general, as it is repeatedly stated that it affects only a subset of genes. And indeed, as suggested by the referee, it is the Sty1 MAP kinase that gives the specificity. At most loci, Sty1 is not present and therefore a classical CTD S2P pattern is observed.

2) A critical element of the model is that double S2P+S5P no longer binds Set1 complex (and/or HDACs). This should be tested physically, maybe by peptide binding.

We now have an experiment using GST-CTD, GST-CTD S5A and GST-CTD S2A in vivo that clearly indicates that Set1 association with the CTD is only favoured by the S5 phosphorylated CTD in the absence of S2P. This experiment would nicely address the issue and complement the mechanistic aspect of the paper by showing that the early S2P phosphorylation interferes with the binding of Set1 to the CTD.

[Editors’ note: further revisions were requested before acceptance.]

Thank you for choosing to send your work entitled “Promoter nucleosome dynamics regulated by signaling through the CTD code” for consideration at eLife. Your submission has been considered by James Manley and the Reviewing Editor who oversaw the initial review.

We remain unconvinced of the general significance of the observations and are still concerned regarding the limited mechanistic insight. However, if you can: 1) show that the phenomena observed (ser-2 phosphorylation) occurs at a significant number of promoters in yeast; and 2) provide more solid insights into the mechanism, then we would be willing to consider a new manuscript on this topic.

I am happy to say that we have addressed these issues in a new version of the manuscript. We report a list of 324 genes that show dynamic changes at the -1 nucleosome in the *rpb1 CTD S2A* mutant. This list is highly enriched in genes we have previously identified as having an early CTD S2P profile (Coudreuse et al., Curr. Biol. 2010), and in genes downregulated in the *CTD S2A* mutant (Coudreuse et al., Curr. Biol. 2010, Schwer et al., PNAS 2012). These data are presented in the subsection headed “CTD S2P regulates promoter nucleosome dynamics at a subset of genes” of the manuscript. They establish that the phenomena we describe occur at a significant number of promoters. We propose that the gene specificity is established by MAP kinase signaling, which is essential to recruit the CTD S2 kinase nearby the promoters of the *ste11* and *inv1* genes that we have analysed in more details. In addition, the number of gene affected may be underestimated by the fact that the large-scale analyses were performed in a single growth condition. Regarding the mechanism, we provide a completely new figure (Figure 4) where the association between the CTD and the H3K4 methylase Set1 is shown in vivo to require CTD S5 phosphorylation in the absence of CTD S2 phosphorylation. These new data support our model where a peak of CTD S2P displaces the Set1 methylase, which relieves its repressing effect.